# Assessment of "Sameness" and/or Differences between Marketed Creams Containing Miconazole Nitrate Using a Discriminatory in vitro Release Testing (IVRT) Method

**Potiwa Purazi [1], Seeprarani Rath [1], Ashmita Ramanah [1] and Isadore Kanfer [1,2],***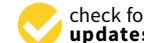

[1]   Biopharmaceutics Research Institute, Rhodes University, Grahamstown 6139, South Africa;
     potiwaprz@gmail.com (P.P.); seeprarath81@gmail.com (S.R.); ashmita.r@hotmail.com (A.R.)
[2]   Leslie Dan Faculty of Pharmacy, University of Toronto, Toronto, ON M5S 3M2, Canada
*   Correspondence: izzy.kanfer@utoronto.ca

**Abstract:** In vitro release testing (IVRT) provides an efficient method for the evaluation of drug release from semi-solid formulations. The aim of this research was to develop and validate a discriminatory IVRT system using vertical diffusion cells (VDCs) to assess generic topical products containing miconazole nitrate (MCZ). A comprehensive approach addressing all essential suitability criteria supporting the reliability of IVRT results was applied. These include mechanical validation of the VDCs, a performance verification test (PVT), validation of the analytical method (HPLC) used to quantify the drug release and validation of the IVRT method to confirm its precision, reproducibility, discriminatory ability, and robustness. Two marketed generic products were tested and assessed in accordance with the acceptance criteria for "sameness" in the FDA's SUPAC-SS guidance which requires that the 90% confidence interval (CI) should fall within the limits of 75%–133.33%. One product was found to be in vitro equivalent to the reference product whereas the other was not. The results confirmed the suitability of the IVRT method to accurately measure the release of MCZ from topical cream products and, importantly, demonstrated the necessary discriminatory ability to assess "sameness"/differences of dermatological creams containing MCZ. Furthermore, the developed IVRT method was able to detect differences between formulations, which may be attributed to qualitative (Q1) and quantitative (Q2) properties and the microstructure and arrangement of matter (Q3).

**Keywords:** IVRT; vertical diffusion cells (VDCs); miconazole nitrate; topical dermatological creams; FDA SUPAC-SS guidance; acceptance criteria; method validation; discriminatory ability; Q1/Q2/Q3 properties

## 1. Introduction

Currently, apart from the use of vasoconstrictor assay (VCA) for the bioequivalence assessment of topical corticosteroid products, comparative clinical trials are the only means whereby a generic company can demonstrate bioequivalence (BE) and/or bioavailability (BA) of most other topical dermatological dosage forms intended for local action [1]. Clinical trials are lengthy, expensive, and require a large number of participants and have resulted in a dearth of generic topical products reaching the market in both developed and developing countries [1,2]. Furthermore, the use of comparative clinical endpoints in patients is considered the least sensitive and reproducible approach among all other approaches to demonstrate BE [3]. Whereas clinical endpoint studies typically utilise a large sample size, they are often insensitive to formulation differences. Hence, faster, less expensive,

and more reproducible and sensitive approaches to reliably demonstrate BE for topical dermatological products are currently being investigated.

In vitro release testing (IVRT) provides a useful method for the evaluation of drug release from semisolid dosage forms and is a valuable tool used in formulation development and as a quality control procedure [1,4,5]. IVRT has been established as a compendial method in the United States Pharmacopeia (USP) General Chapter <1724> where details of the apparatus, procedures for the performance test and statistical methods to assess the "sameness" of such products are described [6].

In this context, "sameness" refers to situations where the qualitative (Q1) and quantitative (Q2) properties, as well as the arrangement of matter (Q3) in the test and reference products, are considered to be the same. In addition, the United States Food and Drug Administration (US FDA) recommends the use of IVRT to assess drug product "sameness" after certain post-approval changes [7], such as manufacturing site or process changes, changes in the components and composition of the drug product, or changes in batch-size. Although IVRT is not generally accepted to establish bioequivalence (BE) and/or bioavailability (BA) for topical drug products, much interest has been directed towards its use as a surrogate measure to obtain a biowaiver. Consequently, the US FDA published draft guidances recommending the use of IVRT for biowaivers of several topical dermatological products [8–11]. More recently, the Committee for Medicinal Products for Human Use (CHMP) of the European Medicines Agency (EMA) has published a draft guideline on quality and equivalence of topical products recommending the application of IVRT [12].

Experts from Fédération Internationale Pharmaceutique/American Association of Pharmaceutical Scientists (FIP/AAPS) suggested that it is crucial to evaluate the influence of each IVRT parameter that may potentially alter the test results, and to validate the operational ranges within which the IVRT remains consistent and reliable for IVRT to attain the status of a generally applicable, robust and valuable tool [13]. In light of the above, the objective of the present investigation was to develop and validate a robust and reproducible IVRT method to confirm "sameness" which has adequate discriminatory power to highlight differences between formulations.

## 2. Materials and Methods

### 2.1. Materials

#### 2.1.1. Chemicals and Formulations

Hydrocortisone (HC), miconazole nitrate (MCZ) and econazole nitrate (ECZ) reference standards were procured from Sigma-Aldrich Co (St. Louis, MO, USA). HPLC-far UV grade acetonitrile was purchased from Microsep (Port Elizabeth, South Africa). HPLC-grade water was prepared by reverse osmosis followed by filtration through a MilliQ system (Millipore®, Bedford, MA, USA). Potassium dihydrogen phosphate orthophosphate and ammonium acetate were purchased from Merck® (Merck®, Wadeville, South Africa).

A marketed HC cream, Emo-Cort 1% (GlaxoSmithKline Inc, Mississauga, ON, Canada), was used for performance verification test (PVT). Daktarin® cream (2%) (Janssen Pharmaceutica, Woodmead, South Africa (Pty) Ltd) was used as the reference product to develop and validate the IVRT method. Three specially manufactured cream formulations containing 1%, 2%, and 4% MCZ, were also used to confirm the sensitivity, specificity, and selectivity of the IVRT method. Two approved and marketed generic formulations, Dermazole® cream (2%) (Sandoz South Africa (Pty) Ltd, Kempton Park, South Africa) and Covarex® cream (2%) (manufactured by MeyerZall Laboratories (Pty) Ltd, George, South Africa, marketed by Glenmark Pharmaceuticals South Africa (Pty) Ltd, Midrand, South Africa) were compared against the reference product to demonstrate the utility of the method to assess "sameness" and/or detect differences.

### 2.1.2. HPLC Instrumentation

The HPLC system used to analyse the PVT and IVRT samples was equipped with a separation module Waters Alliance Model 2695 that consisted of an online degasser module, an autosampler and a photodiode array (PDA) detector (Model 2996- Waters®, Milford, MA, USA). Waters® Empower 3 software (Waters® Milford, MA, USA) was used for data acquisition and processing. A Mettler® Model AE 163 analytical balance (Mettler® Inc., Zurich, Switzerland) was used to weigh the reference standards and creams. Micropipettes P100 and P1000 (Pipetman™, Gilson®, Villiers-le-Bel, France) were used to transfer standard and sample solutions for dilutions.

The HC concentrations in the PVT samples were determined by a validated RP-HPLC method with UV detection at 254 nm using a Phenomenex Luna C18 (2) 5 µm (150 × 4.6 mm) column and a mobile phase consisting of water: acetonitrile (70:30 *v/v*) and a flow rate of 1 mL/min following a 10 µL sample injection.

The MCZ concentrations in the IVRT samples were determined by RP-HPLC using a Phenomenex Luna C18 5 µm (150 × 4.6 mm) column and a mobile phase consisting of ammonium acetate buffer (0.005 M): acetonitrile (adjusted with 0.1% phosphoric acid) (25/75 *v/v*) pumped at a flow rate of 1 mL/min following a 10 µL sample injection. The eluate was monitored at a wavelength of 226 nm and ECZ was used as an internal standard for the determination of MCZ concentrations.

### 2.1.3. IVRT System

The IVRT system used to assess drug release consisted of six vertical diffusion cells (VDCs) (PermeGear, Inc., Hellertown, PA, USA) with a diffusional surface area of 1.767 cm$^2$ mounted on a six-station diffusion apparatus equipped with individual stirrer motors and the cells connected to a Colora® ultra-thermostat water bath/ circulator (Colora®, Lorch, Germany). The donor and receptor chambers were separated by the selected synthetic membrane.

### *2.2. Methods*

### 2.2.1. HPLC Method Validation

The HPLC method for the quantitation of the IVRT samples was validated in accordance with International Council for Harmonization (ICH) guidelines [14] for linearity and range, precision, repeatability, accuracy, specificity, and limit of quantitation (LOQ) and limit of detection (LOD) determined.

### 2.2.2. Mechanical Validation

The capacities and diameters of the VDCs, temperature of the receptor medium, stirring speeds, the dispensed sample volume and the environmental conditions under which the VDC system was placed were assessed as a part of mechanical validation. The parameters were assessed based on the USP Dissolution toolkit procedures for mechanical calibration and PVT for Apparatus 1 and 2 [15–17].

The system was placed on a dedicated, sturdy wooden workbench with a free distance of more than 76 cm above the system. The workbench levelness was measured in two directions with a digital spirit level, ensuring an inclination of less than ± 1°. It was confirmed that the VDC system was not exposed to direct sunlight and not directly exposed to any cooling vents. In addition, failure-resistant power supply to all the electronic components was ensured.

The capacities and the diameters of the orifices of each VDC required for calculation of the amount released per unit area were determined by weight and length measurements, respectively. To assess the capacity of each VDC by weight, the stirring bar and helix were placed into the VDC and the increase in weight was recorded after filling the VDC with purified water. Diameters of the orifice were measured with a Vernier calliper.

The temperature was maintained at 32 ± 0.5 °C to mimic skin temperature and avoid modifications of the diffusion coefficient. The receptor medium temperature was measured in all six VDCs with a calibrated digital thermometer (2006T Type T THERM, Sifam Limited, Torquay Devon, TQ2 7AY England) after the VDCs were filled with purified water and equilibrated for 30 min. The receptor medium was continuously stirred using a magnetic stirrer.

Stirring speed uniformity between the six VDCs was evaluated at 500 rpm using an optical tachometer (Optical tachometer 8 memory version, RS Components, Yokohama, Japan).

All physical parameters were measured in triplicate. Mean ($\bar{x}$) and range of variation (V) were calculated to determine the accuracy and the inter-cell variability of the six VDCs in accordance with predefined acceptance criteria. Acceptance criteria for the apparatus qualification were based on the USP General Chapter <1724> [6,15].

### 2.2.3. Performance Verification Test (PVT) [15,16]

The PVT as described by Hauck et al. [18] was performed using 1% HC cream, Emo-Cort®, and Tuffryn membranes. A calibrated dispenser (Eppendorf Original Model 4780 Repeating Pipette, Eppendorf AG, Hamburg, Germany) was used to transfer ~300 mg of the cream onto the membranes. The amount applied was considered to be sufficient to completely cover the membrane surface in order to maintain an infinite dose in the donor chamber [6,7]. Two VDC runs were conducted on two different days by the same operator using a receptor medium consisting of ethanol: water (30/70 *v/v*). The membranes were presoaked for 30 min in the receptor medium prior to use. The receptor medium was stirred at 500 rpm and the receptor medium temperature was maintained at 32 ± 0.5 °C. The cells were covered with *Parafilm-M* sealing film to prevent evaporation of vehicle and to ensure integrity of the formulations throughout the study. Aliquots of 200 μL were sampled at intervals of 0.5, 1, 2, 3, 4 and 6 hours and replaced with fresh temperature equilibrated receptor fluid to maintain sink conditions. Sample withdrawal and receptor fluid replenishment was achieved using 1mL syringes (Terumo® (Philippines) Corporation, Laguna, Philippines) with sampling needles (21G x 3.5" Terumo® Spinal Needle, Terumo® Corporation, Tokyo, Japan) purchased from local pharmacies.

### 2.2.4. MCZ Solubility in the Receptor Medium

The solubility of MCZ in the receptor medium ethanol: 0.05 M phosphate buffer, pH 4.5 (50/50 *v/v*) was evaluated in triplicate. The drug was weighed and placed into an empty VDC and the receptor medium was added to obtain a saturated concentration. The MCZ solutions were stirred at 32 ± 0.5 °C for 6 h. The stirring was then stopped and samples allowed to stand overnight. Aliquots of the supernatant were withdrawn, filtered, diluted and analysed to determine the concentration of the dissolved MCZ. Ideally, the solubility of the drug should be > 10 times the maximum expected concentration of the drug in the receptor medium throughout the IVRT run [19].

### 2.2.5. Membrane Screening

Membrane screening was carried out using various synthetic membranes such as Magna Nylon (0.45 μm, 25 mm, GVS Life Sciences, Sanford, ME, USA), Tuffryn (0.45 μm, 25 mm, Pall Corporation, Ann Arbor, MN, USA), Cellulose acetate (0.45 μm, 25 mm, Sartorius, Göttingen, Germany), Cellulose Ester (0.45 μm, 47 mm, Advantec MFS, Inc., Pleasanton, CA, USA) and Strat-M (25 mm, Millipore Corporation, Billerica, MA, USA). Binding of MCZ on various membranes was investigated in duplicate by immersing each individual membrane in 10 mL of test solution containing 250 μg/mL of MCZ in the receptor medium of ethanol: 0.05M phosphate buffer, pH 4.5 (50/50 *v/v*) at 32 ± 1 °C for 6 hours. A duplicate set of control MCZ solutions without membranes was evaluated in parallel. In order to determine any decrease in the MCZ content, which may be attributed to binding of the drug to the membrane and/or its stability in the receptor medium, the concentrations of all test solutions were determined using the HPLC method described above. Furthermore, an IVRT run was performed to compare the release from two membranes (Nylon and Tuffryn membranes).

### 2.2.6. IVRT Method

All the IVRT runs were conducted using six VDCs in parallel. The receptor medium was stirred at 500 rpm and the temperature maintained at 32 ± 0.5 °C. The receptor chambers were filled with degassed receptor medium, phosphate buffer: ethanol (50:50 *v/v*) and the VDC system was allowed to equilibrate at 32 ± 1 °C for approximately 30 min. The cream (~300 mg) was dispensed using a calibrated dispenser onto the nylon membranes which had been pre-soaked in the receptor medium for 30 min. Aliquots of 200 µL were withdrawn from the receptor chambers of each of the 6 VDCs at 0.5, 1, 2, 3, 4, 5 and 6 h. The VDCs were subsequently replenished with 200 µL of receptor medium after each withdrawal. The stirring was stopped for a period not exceeding 0.5 min during sample withdrawal and immediately resumed once the aliquots were withdrawn and receptor media was replenished in the VDCs. The aliquots were analysed using the HPLC method described above.

### 2.2.7. Calculation of release rates

The Higuchi model [20,21], which assumes the existence of perfect sink conditions, was used to determine the release rates using Equation (1). Dilution of the receptor medium due to replacement of the sampled amount was taken into account, and the concentrations of MCZ in the receptor medium ($C_n$) at different sampling times were calculated using Equation (1).

$$Q_n = C_n \frac{V_c}{A_c} + \frac{V_s}{A_c} \sum_{i=1}^{n} C_{i-1} \tag{1}$$

where,

$Q_n$ = Amount released at time (*n*) per unit area in µg/cm$^2$
$C_n$ = Concentration of drug in receptor medium at different sampling times (*n*) in µg/cm$^3$
$V_s$ = Volume of the sample in cm$^3$
$V_c$ = Volume of the cell in cm$^3$
$A_c$ = Area of the orifice of the cell in cm$^2$

The release rate corresponds to the slope of the regression line of the plot of $Q_n$ versus square root of time [15].

### 2.2.8. Validation of the IVRT method

- Linearity, Precision, and Reproducibility

Three IVRT runs were conducted using the reference product, Daktarin® cream (2%), with a set of six VDCs on three different days to determine linearity, precision, and reproducibility. In order to comply with Higuchi's assumptions [20,21], the amount released per unit area should be in a linear relationship with the square root of time.

Precision and reproducibility were determined by estimating intra- and inter-run variability from the obtained 18 release rates using Equation (2).

$$\frac{\sigma_1}{\mu} < 0.15 \ \& \ \frac{\sigma_2}{\mu} < 0.15 \tag{2}$$

where,

$\mu$ = Estimated mean release rate
$\sigma_1$ = Inter-run standard deviation
$\sigma_2$ = Intra-run standard deviation

- Sensitivity, Specificity and Selectivity

Sensitivity, specificity and selectivity of the IVRT method were evaluated in three IVRT runs of 6 VDCs each using creams containing 1%, 2%, and 4% MCZ, that were specially formulated and extemporaneously prepared by levigation using aqueous cream BP and the MCZ reference standard.

Sensitivity of the IVRT method was validated by determining changes in the release rates as a function of MCZ concentration.

The specificity was characterised by evaluating the proportionality of the release rates with respect to the MCZ concentration in the test products. A linear regression model with the release rate as dependent variable and MCZ concentration as predictor variable was applied to estimate the coefficient of determination ($R^2$).

In order to test if the IVRT method was selective to accurately identify differences in product performance, the cream containing 2% MCZ was compared to the test products with higher and lower MCZ concentrations using the statistical approach described in the USP General Chapter <1724> [6]. Additionally, the ability of the IVRT method to accurately demonstrate "sameness" was tested by a pairwise comparison of Daktarin® cream (2%) against itself using the results from the three runs performed for linearity, precision and reproducibility.

- Robustness

The robustness of the method to minor perturbations (± 2°C) in nominal temperature (i.e. 32°C) was evaluated at 30°C and 34°C, respectively.

- Recovery

In order to ensure that there was no excessive dose depletion during the IVRT runs, the results from the linearity, precision and reproducibility tests were used to calculate the recovery using Equation (3).

$$Recovery = \frac{Cumulative\ amount\ of\ MCZ\ in\ the\ receptor\ solution\ at\ the\ last\ time\ point}{Strength\ of\ the\ product\ x\ Amount\ of\ applied\ dose} \tag{3}$$

### 2.2.9. Assessment of "Sameness"/Differences between Creams Containing 2% MCZ

The comparative IVRT studies were conducted in accordance with the FDA's SUPAC-SS guidance [7]. The test products, i.e. two approved and marketed generic formulations, Dermazole® cream (2%) and Covarex® cream (2%), were compared against the reference product, Daktarin® cream (2%) as shown in Figure 1. The VDCs were assigned randomly to the test (T) and reference (R) products in accordance with the SUPAC-SS guidance [7]. The individual cumulative amounts of drug released from R and T were plotted versus the square root of time. A nonparametric statistical method, Mann-Whitney U test, was used to calculate the 90% confidence interval (CI) for the ratio of slopes between R and T. Since a few outliers are expected to occur during IVRT (e.g., due to air bubble formation), a nonparametric method that tends to be resistant to the presence of such outliers was used.

**Schematic representation of the IVRT method**

**Figure 1.** Schematic representation of the IVRT method used to assess "sameness"/differences between creams containing 2% MCZ.

### 2.2.10. Application of IVRT

The ability of the IVRT method to detect formulation differences was investigated using two different strengths (0.5% and 1%) of MCZ cream prepared by diluting the Daktarin® cream (2%) with aqueous cream BP. These diluted products were compared against the commercially available Daktarin® cream (2%) product.

## 3. Results

### 3.1. HPLC Method Validation

The results obtained for the HPLC method validation are summarised below:

- **Concentration range -** 5, 10, 20, 40, 60, 80 and 100 µg/mL were used as calibrators
- Linearity - $R^2 \geq 0.999$
- **Accuracy -** CV < 5% using the following concentrations: 20, 100 and 180 µg/mL
- **Repeatability -** CV < 5% for 30, 50, 70 µg/mL during 3 separate runs
- Inter- and intra-run precision - CV < 5%
- **LLOQ and LOD -** 1 and 0.33 µg/mL, respectively
- **Sample stability -** Bench-top (24 ± 1 °C), HPLC sample tray (21 °C ) and refrigerator (4 ± 1 °C) for 9 days
- **Specificity -** The placebo extract did not show any interfering peaks. In addition, the specificity was proved when the average MCZ content in three different marketed products was analysed and found to be between 95.4% and 101.3% of the label claim. CV < 5%.

The validation parameters for the HPLC method complied with the pre-defined acceptance criteria in accordance with the ICH guidelines [14].

### 3.2. Mechanical Validation

The various parameters assessed during mechanical validation of the VDC system complied with the pre-defined acceptance criteria which are summarised in Table 1.

**Table 1.** Pre-defined acceptance criteria and results mechanical validation.

| Test | Min | Max | V | Acceptance Criteria | Pass |
|---|---|---|---|---|---|
| Bench top levelness | - | - | - | not more than 1° | Yes |
| Capacity of the cells | 11.98 mL | 12.04mL | 0.06 mL | $\bar{x}_i \in [12 + 0.1\ mL,\ 12 - 0.1\ mL]$ for $1 \leq i \leq 6$ | Yes |
| Temperature of the receptor medium | 32.0 °C | 32.3 °C | 0.3 °C | $\bar{x}_i \in [32 + 1\ °C, 32 - 1\ °C]$ for $1 \leq i \leq 6$ | Yes |
| Diameter of the orifice of the cell | 14.90 mm | 15. 32 mm | 0.42 mm | $\bar{x}_i \in [15 + 0.55\ mm,\ 15 - 0.55\ mm]$ for $1 \leq i \leq 6$ | Yes |
| Dispensed sampling volume | 200 µL | 200 µL | - | $\bar{x}_i \in [200 + 15\ µL,\ 200 - 15\ µL]$ for $1 \leq i \leq 6$ | Yes |
| Stirring speed | 499.6 rpm | 500.2 rpm | 0.6 | $\bar{x}_i \in [500 + 50\ rpm,\ 500 - 50\ rpm]$ for $1 \leq i \leq 6$ | Yes |

### 3.3. Performance Verification Test (PVT)

The PVT runs were in agreement with the pre-defined acceptance criteria and the results are shown in Table 2. The resulting data confirm the suitability and reproducibility of the IVRT system.

**Table 2.** Pre-defined acceptance criteria and results for the performance verification test (PVT).

| Parameter | Acceptance Criteria | Results |
|---|---|---|
| Intra-run variability | CV for the first run (*n* = 6 VDCs) <15% | 3.01% |
| | CV for the second run (*n* = 6 VDCs) <15% | 11.90% |
| Inter-run variability | CV for both runs (*n* = 12 VDCs) <15% | 8.87% |
| Product "sameness" testing | The 90% CI must fall within the limits of 75–133.33% | Lower limit: 88.41% Upper limit: 113.37% |

### 3.4. MCZ Solubility in the Receptor Medium

The receptor medium of 50% ethanol and 50% of 0.05M phosphate buffer, pH 4.5 was chosen after some preliminary experiments with different pH and molarity of the buffer. Ethanol: 0.05M phosphate buffer pH 4.5 (50/50 *v/v*) was chosen as receptor medium for IVRT since MCZ was highly soluble in this composition. The solubility of MCZ in the receptor medium was 3671 µg/mL (±60.24) which was more than 10 times higher than the highest measured concentration in the samples obtained during the method validation experiment (150 µg/mL). The results confirmed that the solubility of MCZ in the receptor medium was sufficiently high to prevent saturation effects and to ensure sink conditions [19].

### 3.5. Membrane Screening

The nylon, cellulose acetate and Tuffryn membranes showed average percentage recoveries of 98.40%, 95.10%, and 98.45%, respectively indicating negligible MCZ binding. The membranes, therefore, are unlikely to act as a rate limiting barrier for MCZ. Contrarily, Strat M and cellulose ester membranes exhibited low recoveries of MCZ and did not conform to the acceptable range of ± 10 %, which may be attributed to MCZ binding.

A comparison of the in vitro release of MCZ from Daktarin cream (2%) using nylon and Tuffryn membranes after 6 h showed release rates of 86.64 µg/cm$^2$/min$^{1/2}$ and 76.68 µg/cm$^2$/min$^{1/2}$. Subsequently, nylon was chosen due to higher release rate, availability, and affordability.

### 3.6. Validation of the IVRT Method

- Linearity, Precision, and Reproducibility

Linearity is confirmed when the R$^2$ > 0.9. The 18 resulting release rates from 3 runs (n = 6) were calculated using linear regression according to the SUPAC-SS guidance [7]. All release rates showed a

linear relationship ($R^2 > 0.99$) between the amount of drug released into the receptor medium per unit area ($\mu g/cm^2$) and the square root of time.

The variability parameters were equal to $\sigma_1^2 = 36.14$, $\sigma_2^2 = 27.56$, and $\mu = 57.86$ $\mu g/cm^2/min^{1/2}$, resulting in a CV of 10.39% for the inter-run variability and a CV of 9.07% for the intra-run variability. CVs < 15% confirm acceptable precision and reproducibility.

- Sensitivity

The mean release rates increased with an increase in MCZ concentration: 14.56 $\mu g/cm^2/min^{1/2}$ (±4.84), 39.92 $\mu g/cm^2/min^{1/2}$ (±6.00) and 73.55 $\mu g/cm^2/min^{1/2}$ (±3.78) for the creams containing 1%, 2%, and 4% MCZ, respectively. The method was considered to be sensitive since the cream containing 1% MCZ yielded a lower mean release rate than that containing 2% MCZ, whereas the cream containing 4% MCZ yielded the highest mean release rate.

- Specificity

A linear relationship ($R^2 = 0.9935$) between the MCZ concentration and the release rates was observed (Figure 2). Hence, the specificity of the IVRT method was confirmed since the mean release rates of the test products increased proportionately with the MCZ concentration.

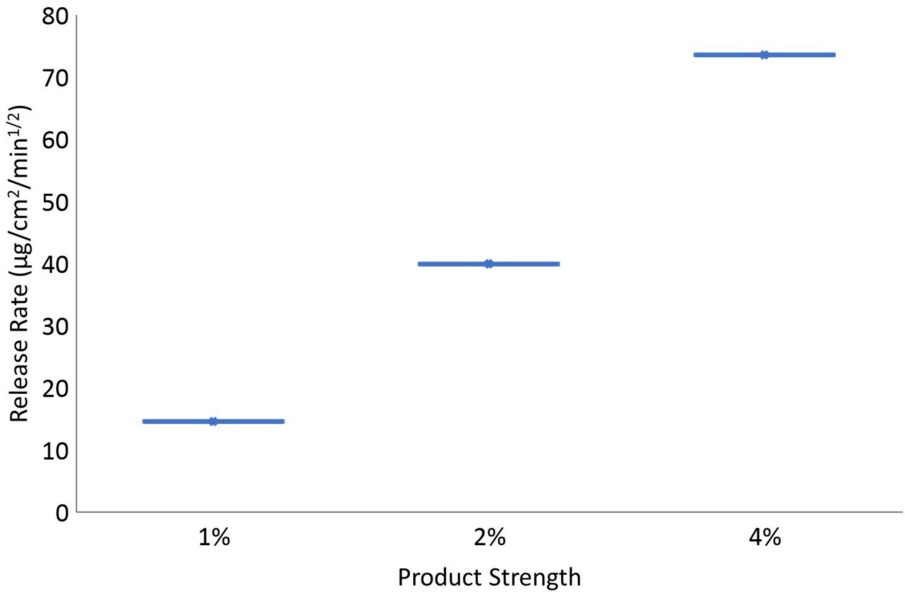

**Figure 2.** Box and Whiskers plot of the measured release rates for the three products containing 1%, 2%, and 4% miconazole nitrate (MCZ).

- Selectivity

Pairwise comparison of the 2% MCZ cream with the 1% and 4% MCZ creams indicates the ability to accurately identify inequivalent products whereas the comparison of the reference product with itself confirm the determination of "sameness" (Table 3).

- Robustness

The mean release rates from the IVRT runs using different temperatures i.e. 30 °C and 34 °C were 55.14 $\mu g/cm^2/min^{1/2}$ (±8.15) and 72.53 $\mu g/cm^2/min^{1/2}$ (±9.00), respectively. The mean release rates did not deviate by > 15% from that obtained from the IVRT runs performed at nominal temperature (32 °C) i.e., 59.30 $\mu g/cm^2/min^{1/2}$ (±7.27), hence confirming the robustness of the developed IVRT method to minor perturbations in temperature during experimental runs.

**Table 3.** Computed 90% confidence interval (CI) for the ratio of the release rate between two runs.

| Pairwise Comparison | Computed 90% CI | | "Sameness" Confirmed |
| --- | --- | --- | --- |
| | Lower Limit | Upper Limit | |
| MCZ cream 1% *vs* MCZ cream 2% | 25.25 | 51.65 | No |
| MCZ cream 4% *vs* MCZ cream 2% | 165.43 | 210.76 | No |
| Daktarin® cream (2%) (Run 1) *vs* Daktarin® cream (2%) (Run 2) | 83.39 | 104.20 | Yes |
| Daktarin® cream (2%) (Run 1) *vs* Daktarin® cream (2%) (Run 3) | 88.88 | 112.88 | Yes |
| Daktarin® cream (2%) (Run 2) *vs* Daktarin® cream (2%) (Run 3) | 93.29 | 122.03 | Yes |

- Recovery

The recoveries observed during the IVRT runs performed to test linearity, precision and reproducibility were < 30% i.e., 27.11% (±2.00), 28.20% (±2.41) and 29.75% (±3.26), respectively, indicating that any dose depletion is unlikely within the established experimental conditions as there was no significant curvature in the release profiles.

### 3.7. Assessment of "Sameness"/Differences between Creams Containing 2% MCZ

The release profiles and the pairwise comparison of the reference product, Daktarin cream (2%), against two approved and marketed generic formulations, Dermazole® cream (2%) and Covarex® cream (2%) are shown in Figure 3 and Table 4, respectively.

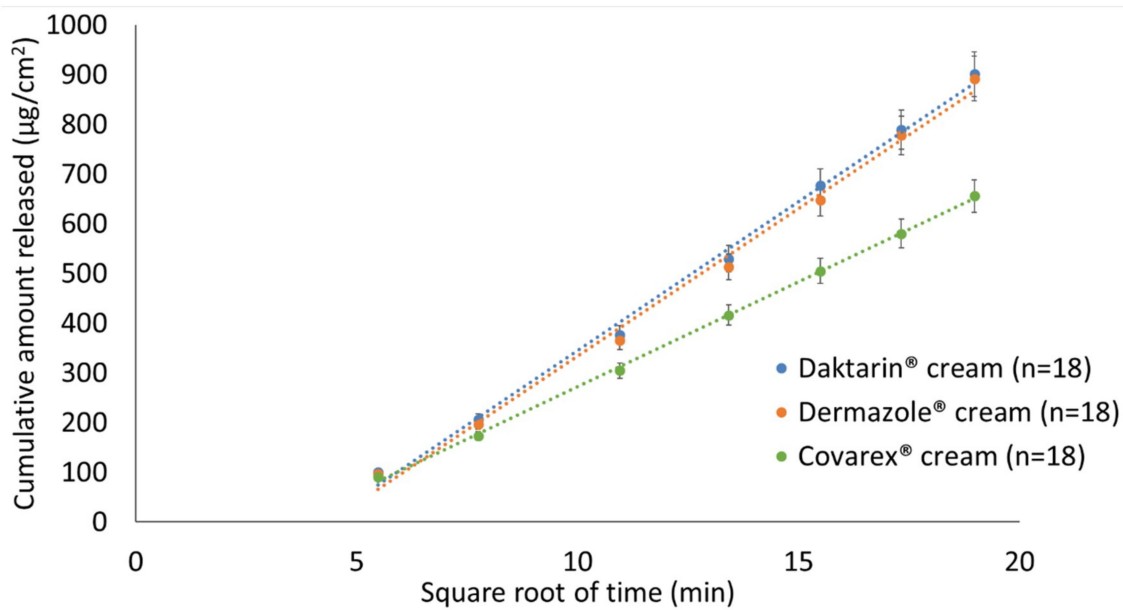

**Figure 3.** Comparison of the release profiles of marketed creams, Dermazole® cream (2%) and Covarex® cream (2%) against the reference product, Daktarin cream (2%).

**Table 4.** Computed 90% CI for the comparative in vitro release testing (IVRT) of the marketed generic products against the reference product.

| Pairwise Comparison | Computed 90% CI | | "Sameness" Confirmed |
|---|---|---|---|
| | **Lower Limit** | **Upper Limit** | |
| Daktarin® Cream (2%) *vs* Dermazole® cream (2%) | 93.93 | 103.74 | Yes |
| Daktarin® cream (2%) *vs* Covarex® cream (2%) | 61.93 | 72.98 | No |
| Dermazole® cream (2%) *vs* Covarex® Cream (2%) | 68.26 | 74.50 | No |
| Daktarin® cream (2%) *vs* Daktarin® cream (2%) | 97.10 | 113.39 | Yes |

The 90% CI for Dermazole® cream (2%) when compared against the reference product, Daktarin cream (2%) fell within the limits of the SUPAC-SS acceptance criteria (75–133.33%) confirming "sameness" between the two creams whereas, the 90% CI for Covarex® cream (2%) when compared against the reference product, Daktarin cream (2%), as well as the generic product, Dermazole® cream (2%) fell completely outside the acceptance limits indicating that the creams were not pharmaceutically equivalent.

The difference in performance criteria of topical dosage forms can be explained by considering Q1, Q2 and Q3 which refer to the qualitative and quantitative properties and formulation microstructure, respectively [22]. The difference in drug release profiles observed with Covarex® cream may be due to differences in excipients (Q1/Q2). Covarex® cream contained imidazolidinyl urea (0.2%), methyl paraben (0.15%) and sodium propyl paraben (0.15%) whilst Daktarin® and Dermazole® creams contained benzoic acid (0.2%). Such differences in excipients can alter the physicochemical properties, skin permeation, solubility and thermodynamic activity of the drug formulation as well as the Q3 attributes of the formulation [22].

The results of the investigation to detect formulation differences as a result of differences in method of preparation are shown in Figure 4.

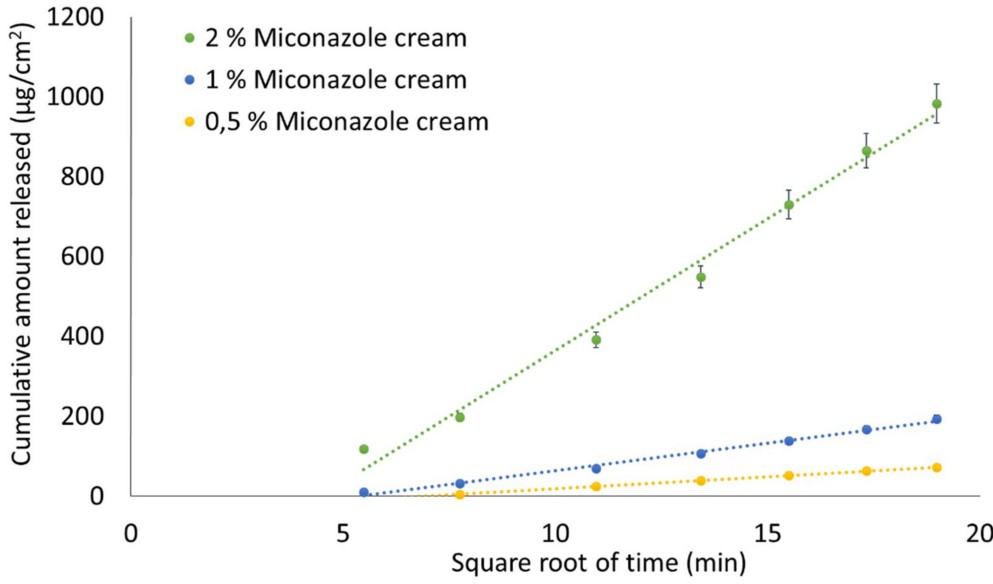

**Figure 4.** Release profiles of creams containing 0.5%, 1%, and 2% MCZ prepared by dilution method.

The rate of release of MCZ from Daktarin® cream (2%) was markedly higher than the 1% and 0.5% MCZ cream formulations (Figure 4). This is probably due to Q3 differences, i.e. related to the

difference in the microstructure and arrangement of matter [22] as well as differences in Q1/Q2 as a result of the dilution with aqueous cream BP. The release profiles shown in Figure 4 indicate that the IVRT system can differentiate formulations which have been formulated using different methods.

## 4. Conclusions

A comprehensive approach addressing all qualification/validation considerations of an IVRT method for topical MCZ cream was carried out.

In vitro evaluations of approved and marketed topical MCZ creams were carried out according to recommendations of the US FDA Draft Guidance for acyclovir ointment [8] and cream [10] as well as the SUPAC SS Guidance for non-sterile semisolid dosage forms [7] and the USP General Chapter <1724> [6]. A generic cream, Dermazole® cream (2%), showed "sameness" to the reference product: Daktarin® cream (2%), whereas another generic cream, Covarex® cream (2%), whose MCZ release rate did not fall within the acceptance criteria of 75–133.33% was considered not to be in vitro equivalent. The comparison between the two generic creams indicated that these were not in vitro equivalent.

In summary, the IVRT system developed illustrated the additional important and critical properties of being able to accurately discriminate between in vitro release rates, which could reflect the similarity and/or differences in product performance. Furthermore, the results indicate that the developed IVRT method has the valuable ability to also detect changes in a formulation, which may be due to Q1/Q2/Q3 properties. The validated IVRT system provides evidence that the experimental design, equipment, and methodology has the necessary requirements to accurately evaluate the release of MCZ from topical cream products for both innovator and generic formulations. Results confirm the suitability of the IVRT method to measure the release rate of MCZ from topical dermatological creams in a reliable and reproducible manner, and provides compelling data for future use to obtain biowaivers for such products.

**Author Contributions:** Conceptualization, I.K.; Funding acquisition, I.K.; Investigation, P.P., S.R. and I.K.; Methodology, P.P., S.R. and I.K.; Resources, I.K.; Supervision, I.K.; Validation, P.P., S.R., A.R. and I.K.; Visualization, S.R.; Writing—original draft, P.P.; Writing—review & editing, S.R., A.R. and I.K. All authors have read and agreed to the published version of the manuscript.

**Funding:** This research was funded by the Biopharmaceutics Research Institute, Rhodes University, Grahamstown, South Africa.

**Conflicts of Interest:** The authors declare no conflict of interest.

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
