# Peer review of "Assessment of “Sameness” and/or Differences between Marketed Creams Containing Miconazole Nitrate Using a Discriminatory in vitro Release Testing (IVRT) Method"

_scipharm, doi:10.3390/scipharm88010006_

Round 1
Reviewer 1 Report
Dear authors,
Your manuscript fits the scope of the journal. It is well written, presented, and the conclusions are supported by the data. The methodology and the study design are fine.
Nevertheless, prior to publication, please make the following minor corrections/implementation:
1- Abstract: statistical means are lacking; Also, please define Q1/Q2/Q3
2- Key References to add:
a-Guimaraes, C. , Menaa, F. , Menaa, B. , Lebrun, I. , Quenca-Guillen, J. , Auada, A. , Mercuri, L. , Ferreira, P. and Santoro, M. (2010) Determination of isotretinoin in pharmaceutical formulations by reversed-phase HPLC. Journal of Biomedical Science and Engineering, 3, 454-458. doi: 10.4236/jbise.2010.35063.
b-
Thermochimica Acta
Volume 505, Issues 1–2, 10 June 2010, Pages 73-78
Comparative physical–chemical characterization of encapsulated lipid-based isotretinoin products assessed by particle size distribution and thermal behavior analyses
Author links open overlay panelCarla AiolfiGuimarãesaMaria Inês Rocha MiritelloSantoroa
Author Response
Comment 1 - Abstract: statistical means are lacking; Also, please define Q1/Q2/Q3
Response - The following sentence regarding the statistical means has been added on lines 18-21.
“Two marketed generic products were tested and assessed in accordance with the acceptance criteria for “sameness” in the FDA’s SUPAC-SS guidance which requires that the 90% confidence interval (CI) should fall within the limits of 75%–133.33%.”
Q1/Q2/Q3 is now defined in the abstract on lines 26-27 as follows:
“qualitative (Q1) and quantitative (Q2) properties and the microstructure and arrangement of matter (Q3)”.
Comment 2 - Key References to add:
a. Guimaraes, C. , Menaa, F. , Menaa, B. , Lebrun, I. , Quenca-Guillen, J. , Auada, A. , Mercuri, L. , Ferreira, P. and Santoro, M. (2010) Determination of isotretinoin in pharmaceutical formulations by reversed-phase HPLC. Journal of Biomedical Science and Engineering, 3, 454-458. doi: 10.4236/jbise.2010.35063.
b. Thermochimica Acta
Volume 505, Issues 1–2, 10 June 2010, Pages 73-78
Comparative physical–chemical characterization of encapsulated lipid-based isotretinoin products assessed by particle size distribution and thermal behavior analyses
Carla AiolfiGuimarãesaMaria Inês Rocha MiritelloSantoroa
Response - Whereas we are prepared to include the two references requested by Reviewer 1, it would be appreciated if the appropriate position for citation of these references could be provided in terms of their relevancy.
Reviewer 2 Report
Were there any other receptor media tested before finalizing the selected receptor? IVRT can be a partitioning study based on the receptor The amount of product applied chosen was on the higher side (300mg). Would the authors chose to comment in the relevant section?
Author Response
Comment 1 - Were there any other receptor media tested before finalizing the selected receptor?
Response - A mixture of 50% ethanol and phosphate buffer was initially chosen on the basis of the solubility of MCZ and tested using molarities of 0.04 M, 0.05 M and 0.06 M phosphate buffer and pHs of 4.5 and 5.5 for each mixture. It was found that the receptor medium of 50% ethanol and 50% of 0.05M phosphate buffer, pH 4.5 was preferable and provided a higher solubility in comparison to the others.
Comment 2 - The amount of product applied chosen was on the higher side (300mg). Would the authors chose to comment in the relevant section?
Response - The following has been added on lines 152-153 regarding the amount of product applied:
“The amount applied was considered to be sufficient to completely cover the membrane surface in order to maintain an infinite dose in the donor chamber [6,7].”
Reviewer 3 Report
Manuscript scipharm-644041 reports a well-done study and is of great interest. Manuscript has a correct structure and includes clear descriptions and explanations of the experimental design. The manuscript has valuable data. It presents data in an easy to follow way. The analytical approach is appropriate and the conclusions are coherent with the data and give some perspective for future studies. The style and the English language are fine, text requires a minor spell check.
- Line 37, please define “BE”
- Line 41, please define “IVRT”
- Line 56, define “FIP/AAPS”
- Include in the introduction a suitable definition of identity
- Line 64, define “HT”
- Line 87, define “PDA”
- Line 110, define “ICH”, “LOQ” and “LOD”
- Please add a schematic representation of the experimental setup
Author Response
Comment 1 - Line 37, please define “BE”
Response - Inserted on Line 35, bioequivalence (BE)
Comment 2 - Line 41, please define “IVRT”
Response - Inserted on Line 47, In vitro release testing (IVRT)
Comment 3 - Line 56, define “FIP/AAPS”
Response - Inserted on Line 56, Fédération Internationale Pharmaceutique/American Association of Pharmaceutical Scientists (FIP/AAPS)
Comment 4 - Include in the introduction a suitable definition of identity
Response - Please clarify and advise intention.
Comment 5 - Line 64, define “HT”
Response - Regret, unable to locate definition of HT. Should we remove “HT” and refer only to “Tuffryn”?
Comment 6 - Line 87, define “PDA”
Response - Inserted on Line 94, photodiode array (PDA)
Comment 7 - Line 110, define “ICH”, “LOQ” and “LOD”
Response - Inserted on Lines 118-119, International Council for Harmonization (ICH) guidelines, limit of quantitation (LOQ) and limit of detection (LOD)
Comment 8 - Please add a schematic representation of the experimental setup
Response - Please advise where such a schematic representation should be located, i.e. as a separate figure or embedded in the text of the manuscript. An example of such a schematic will be much appreciated.
Round 2
Reviewer 3 Report
Include in the introduction a suitable definition of "sameness”.
A schematic representation should be located in the text of the manuscript.
Author Response
The definition of "sameness" has been added on Lines 52-54 as follows :
In this context, “sameness” refers to situations where the qualitative (Q1) and quantitative (Q2) properties as well as the arrangement of matter (Q3) in the test and reference products are considered to be the same.
The following schematic representation of the IVRT method has been added in section 2.2.9.